# Genetic Landscape of Chronic Myeloid Leukemia and a Novel Targeted Drug for Overcoming Resistance

**DOI:** 10.3390/ijms241813806

**Published:** 2023-09-07

**Authors:** Ryo Yoshimaru, Yosuke Minami

**Affiliations:** Department of Hematology, National Cancer Center Hospital East, 6-5-1 Kashiwanoha, Kashiwa-shi 277-8577, Japan; ryyoshim@east.ncc.go.jp

**Keywords:** chronic myeloid leukemia, genome profiling, asciminib

## Abstract

Tyrosine kinase inhibitors (TKIs) exemplify the success of molecular targeted therapy for chronic myeloid leukemia (CML). However, some patients do not respond to TKI therapy. Mutations in the kinase domain of *BCR::ABL1* are the most extensively studied mechanism of TKI resistance in CML, but *BCR::ABL1*-independent mechanisms are involved in some cases. There are two known types of mechanisms that contribute to resistance: mutations in known cancer-related genes; and Philadelphia-associated rearrangements, a novel mechanism of genomic heterogeneity that occurs at the time of the Philadelphia chromosome formation. Most chronic-phase and accelerated-phase CML patients who were treated with the third-generation TKI for drug resistance harbored one or more cancer gene mutations. Cancer gene mutations and additional chromosomal abnormalities were found to be independently associated with progression-free survival. The novel agent asciminib specifically inhibits the ABL myristoyl pocket (STAMP) and shows better efficacy and less toxicity than other TKIs due to its high target specificity. In the future, pooled analyses of various studies should address whether additional genetic analyses could guide risk-adapted therapy and lead to a final cure for CML.

## 1. Introduction

Chronic myeloid leukemia (CML) is a myeloproliferative neoplasm characterized by a marked increase in myeloid cells, and it requires a genetically based diagnosis and management. CML is caused by the reciprocal translocation, t(9;22)(q34;q11), of chromosomes 9 and 22, the Philadelphia (Ph) chromosome, in pluripotent hematopoietic stem cells. This reciprocal translocation translocates the intracellular Abelson murine leukemia (ABL) tyrosine kinase gene on chromosome 9 to the breakpoint cluster region (BCR) gene on chromosome 22 and produces *BCR::ABL* chimeric mRNA. The BCR::ABL protein is formed as a fusion gene product of the Ph chromosome, and it permanently activates tyrosine kinase. BCR::ABL tyrosine kinase auto-phosphorylates by forming multimeric complexes, leading to the phosphorylation of intracellular substrates involved in signaling, while increasing enzymatic activity. As a result, it contributes to leukemic cell growth, proliferation, and survival by suppressing apoptosis.

Tyrosine kinase inhibitors (TKIs) bind to the adenosine triphosphate (ATP)-binding domain of BCR::ABL and block the leukemogenic signaling of BCR::ABL by inhibiting the phosphorylation of the substrates.

TKIs exemplify the success of molecular targeted therapy, as they have enabled the average life expectancy of patients with CML to approach that of the general population [1]. However, some patients fail to achieve the time-dependent endpoints of a complete hematologic response (CHR: the normalization of peripheral blood counts and resolution of splenomegaly and CML-related symptoms), complete cytogenetic response (CCyR: 0% Ph+ metaphases based on the analysis of 20 bone marrow cells), and major molecular response (MMR: *BCR::ABL1* ≤ 0.1% on the international scale (IS)) (primary resistance), or lose response (secondary resistance), or experience intolerance [2,3]. During the 10-year follow-up period in the IRIS study (Phase III clinical trial comparing imatinib alone vs. interferon alpha plus cytarabine), 183 (33.1%) of the 553 patients in the imatinib arm discontinued therapy due to adverse events (*N* = 37, 6.7%), unsatisfactory therapeutic effects (*N* = 29, 5.2%), or death (*N* = 2, 0.4%) [4]. In the DASISION study (Phase III clinical trial comparing imatinib vs. dasatinib), 100 (38.8%) of the 258 patients in the dasatinib arm discontinued treatment due to treatment-related or -unrelated toxicities (*N* = 64, 21%), or progression or treatment failure (*N* = 28, 11%) [5]. During the 10-year follow-up period in the ENEST trial (Phase III clinical trial comparing imatinib vs. nilotinib), 175 (62.1%) of the 282 patients discontinued treatment due to adverse events (*N* = 53, 18.8%), a suboptimal response or treatment failure (*N* = 37, 13.1%), disease progression to the accelerated phase or blast crisis (AP/BC) (*N* = 2, 0.7%), or death (*N* = 9, 3.2%) [6]. In the BFORE trial (Phase III clinical trial comparing imatinib vs. bosutinib), 108 (40.3%) of the 268 patients discontinued treatment due to adverse events (*N* = 67, 25%), a suboptimal response or treatment failure (*N* = 13, 4.8%), disease progression to AP/BC (*N* = 2, 0.7%), or death (*N* = 3, 1.1%) [7]. Although the EPIC trial (Phase III clinical trial comparing imatinib vs. ponatinib) was terminated early following concerns about vascular adverse events observed in patients given ponatinib in other trials, 11 (7%) of the 154 patients in the ponatinib arm had discontinued treatment due to adverse events that were considered possibly related to the study treatment [8]. It is estimated that over 25% of CML patients will switch TKIs at least once during their lifetime due to TKI intolerance or resistance. Mutations in the kinase domain of *BCR::ABL1* are the most extensively studied mechanism of TKI resistance in CML, but they fail to explain approximately 20% to 40% of resistant cases [2]. TKI resistance mechanisms are usually subdivided into *BCR::ABL1*-dependent and -independent mechanisms [9]. These resistant cases may involve *BCR::ABL1*-independent survival pathways [2].

While *BCR::ABL1* kinase domain mutations remain the major known mechanism of acquired TKI resistance, the events leading to the initiation of *BCR::ABL1*-independent resistance remain poorly understood [10]. Several mechanisms are thought to be associated with TKI resistance, including BCR::ABL1 overexpression, abnormal activity of drug transporters, activation of alternative signaling pathways, DNA repair, genomic instability, epigenetic dysfunction, leukemia stem cell (LSC) persistence, and dysfunction of the immune system [11]. Genomic mutations and sequence rearrangements have been reported as mechanisms that may affect the treatment response and contribute to drug resistance and disease progression [12].

Kinase domain mutations confer resistance to TKIs by interfering with the binding of TKIs to the ATP-binding site of BCR::ABL1 [12,13]. *BCR::ABL1* mutation analysis is now routinely performed to monitor patients with treatment failure. The selection of TKIs is based on the TKI sensitivity profiles of the various mutations [12]. In particular, the T315I residue is predicted to confer resistance and is associated with a significantly worse prognosis compared to other mutations [12,14]. Mutations located in the P-loop of *BCR::ABL1* are associated with poor outcomes and more advanced disease [13,15] (Figure 1). Patients with multiple mutations have a poorer response to nilotinib/dasatinib, and patients with T315I plus additional mutations have a poorer response to ponatinib, even though ponatinib is generally effective against most resistance mutations, including the T315I mutation [16]. We will also explain resistance mechanisms other than mutations in the kinase domain. BCR::ABL1 expression due to gene amplification or upregulation at the transcriptional level is another resistance mechanism. The overexpression of BCR::ABL1 leads to resistance by increasing the oncoprotein concentration needed to be inhibited with TKI [17]. Some studies comparing sensitive and resistant imatinib CML patients demonstrated that patients resistant to therapy have higher expression levels of DNA damage repair genes such as *RAD51L1*, *FANCA*, and *ERCC5* [18,19]. These facts support that DNA damage repair impairment in CML is directly involved in TKI resistance and CML evolution. The balance between drug influx and drug efflux is crucial to BCR::ABL1 inhibition by TKIs, and changes in these transporters may explain the resistance phenotypes caused by ineffective TKI uptake and/or excessive extrusion of TKI from the cell. The roles of ABCB1, ABCG2, SLCO1B, and others have been investigated, and transmembrane transporters are considered pivotal because they affect both intracellular and plasma concentrations of drugs. Concentrations of imatinib, nilotinib, and other TKIs may be altered, leading to significant variations in efficacy and tolerability [20]. To overcome the inhibition of BCR::ABL1, CML cells may activate alternative signaling pathways to compensate for the loss of BCR::ABL1 kinase activity. Consequently, the cells will be able to proliferate and survive despite effective BCR::ABL1 inhibition. RAS/MAPK, SRC, JAK/STAT, and PI3K/AKT are some of the pathways that contribute to TKI resistance [11]. In CML, risk scores, such as Sokal, Hasford, EUTOS, and ELTS, are based on the clinical parameters at the time of diagnosis and can stratify patients according to their long-term prognosis. However, there is a lack of a comprehensive and systematic method for evaluating the mutation landscape at the time of CML diagnosis and blast crisis (BC), and for assessing TKI resistance. In addition to the clinical scoring systems, the screening of additional genes at the time of diagnosis may be useful for guiding therapeutic decisions. Resistance-associated mutations that have been detected by next-generation sequencing (NGS) are clinically useful since the early detection of such mutations can aid in guiding treatment decisions and interventions to ensure that the appropriate TKIs are selected to prevent clonal expansion [21].

TKIs that can overcome the toxicity associated with second- and third-generation TKIs are currently under development.

## 2. Rearrangements Associated with the Formation of the Ph Chromosome

When the *BCR::ABL1* fusion is formed, genetic variations occur in the breakpoints involved in initiating the translocation on chromosomes 9 and 22. Furthermore, additional genomic rearrangements accompanying the formation of the *BCR::ABL1* gene fusion are also a source of genetic heterogeneity in CML (Figure 2). A sequence deletion was revealed to be an independent predictor of poor patient outcomes. Cytogenetic and fluorescence in situ hybridization analyses are used in the management of CML, particularly for diagnosis and resistance assessment [12]. The most common breakpoint in CML occurs in the 5.8 kb major breakpoint region of *BCR*, and the 140 kb region between exons 1b and 2 of *ABL1*. This results in a 210 kDa *BCR::ABL1* protein, and common *BCR::ABL1* transcripts in which either *BCR* exon 13 or 14 is fused to *ABL1* exon 2. These common *BCR::ABL1* transcripts are found in >90% of CML patients. However, in some patients, other types of transcripts are found; these are defined as rare or atypical transcripts [22,23]. There have been conflicting opinions as to whether the type of genetic variation in the transcripts affects the treatment outcome. Thus, current treatment guidelines do not consider the transcript type for treatment decisions. Nonetheless, various studies have reported that the *BCR::ABL* transcript type does affect the treatment outcome, and may be an independent predictor of sustained treatment-free remission [24]. Therefore, testing for transcript types may be clinically useful in determining the probability of achieving treatment-free remission.

The *BCR::ABL1* fusion is formed when the terminal sequence of the q arm of chromosome 9 changes place with the terminal sequence of the q arm of chromosome 22. Genetic events involving chromosomes other than chromosomes 9 and 22 in the formation of the *BCR::ABL1* fusion are termed variant translocations [12]; these occur in 5% to 10% of newly diagnosed CML patients [25]. Although it has been reported that patients with variant translocations have inferior outcomes, studies in multiple international cohorts of imatinib-treated patients have found no significant differences in the molecular outcomes, cytogenetic responses, or overall survival between patients with variant or classical translocations [26,27]. Due to these conflicting findings, it remains unclear as to whether variant translocations impact treatment outcomes.

In patients with Ph-associated rearrangements, the *BCR::ABL1* fusions are present as usual, but the *ABL1::BCR* fusions are absent. Instead, *BCR* or *ABL1* is fused to other genes on chromosome 9 or 22, or genes adjacent to *BCR* or *ABL1* form gene fusions. Sequence deletions and inversions are also evident. Such rearrangements are detectable at the time of diagnosis, indicating that they likely occurred at the time of formation of the Ph chromosome and did not emerge at BC [12]. Ph-associated rearrangements are associated with a poor outcome and are more common in cases that progress to the blast crisis [10]. Studies are currently underway to determine whether Ph-associated rearrangements can serve as prognostic markers and guide up-front treatment decisions.

## 3. Pathogenic Genomic Alterations in Hematologic Malignancies

NGS studies have confirmed the role of mutations in cancer-related genes other than *BCR::ABL1* mutations in drug resistance and treatment failure. Seven genes (*ASXL1*, *RUNX1*, *IKZF1*, *BCORL1*, *KMT2D*, *DNMT3A*, *TET2*, *JAK2*, and *TP53*) were recurrently found to be mutated in more than one study, whereas mutated *IDH1* and *IDH2* were rare [28].

Branford et al. [10] found that a cancer gene mutation was detectable at the time of diagnosis in 50% of the 46 patients treated with first-line TKIs; a cancer gene mutation was found in 19 (70%) of the 27 patients with a poor outcome, and in 4 (21%) of the 19 patients with an MMR. Among the 27 patients with a poor outcome, the most frequently mutated gene at the time of diagnosis was *ASXL1* [10]. *ASXL1* mutations cause transcriptional abnormalities due to abnormal chromatin modifications. Patients with mutated *ASXL1* at the time of diagnosis had a significantly longer time to BC than patients with mutations in genes other than *ASXL* at the time of diagnosis [10]. Zhang et al. [29] found that 163 of 169 chronic-phase and accelerated-phase CML patients who were treated with the third-generation TKI for drug resistance had one or more cancer gene mutations. Mutated *ASXL1* was found in 69% of the patients. Furthermore, cancer gene mutations and additional chromosomal abnormalities were independently associated with progression-free survival [29]. Complementary to these findings, Xue et al. found that although cancer gene mutations were detectable in all disease phases, they were detected at the highest frequency (≥85%) in the AP and BC. *ASXL1* mutations were the most common mutations in each disease phase [30]. Patients carrying an *ASXL1* mutation at diagnosis showed a less favorable molecular response to nilotinib treatment, as a major molecular response (MMR) was achieved less frequently at months 12, 18, and 24 compared to all other patients. Patients with *ASXL1* mutations were also younger and more frequently found in the high-risk category, suggesting a central role of clonal evolution associated with *ASXL1* mutations in CML pathogenesis [31].

*RUNX1* is a regulator of hematopoietic cell differentiation. In early studies, it was found to be mutated in BC CML. *RUNX1* mutations contribute to the downregulation of DNA repair machinery and promote genomic instability [32]. Although mutated *RUNX1* is a frequent and aggressive driver of hematologic malignancies, as found in multiple CML studies, it is not always associated with a poor outcome, and it is very rare [10].

In leukemia, *BCOR* and *BCORL1* mutations disrupt the repressive function of PRC1.1 on target genes, resulting in epigenetic reprogramming and activation of aberrant cell signaling programs that mediate treatment resistance. *BCOR* and *BCORL1* mutations are more common in AML than in CML. *BCOR* mutations have been linked to advanced disease and clinical resistance to BCR::ABL-targeted inhibitors [33].

The role of *EZH2* in CML remains unclear, and unlike other myeloid malignancies, inactivating mutations of *EZH2* are rare in CML, although the upregulation of *EZH2* can result in a form of myeloproliferative disease [34].

The United States-based National Comprehensive Cancer Network Clinical Practice Guidelines in Oncology for CML suggest that NGS with a myeloid mutation panel should be considered for patients who present at an advanced disease stage or have disease progression to the advanced phases with no identifiable *BCR::ABL1* kinase domain mutation. BC is the terminal phase of CML that is characterized by the rapid expansion of myeloid or lymphoid differentiation-arrested blast cells, leading to a short median survival. The blast lineage is myeloid in approximately 70% of the cases, and lymphoid in approximately 20% to 30% of the cases. However, lymphoid gene mutations may also be relevant for lymphoid phenotype BC CML, and *BCR::ABL1* mutations frequently co-occur with cancer gene mutations [12,35]. Approximately 85% to 100% of the patients in BC with *BCR::ABL1* kinase domain mutations also had cancer gene mutations, including gene fusions and deletions [12,35,36]. Furthermore, deletions involving the *IKZF1* gene are among the most frequently detected mutations in lymphoid BC [35,36]. Therefore, mutation analyses restricted to single-nucleotide variants, small insertions, and deletions in myeloid genes may fail to detect a substantial proportion of relevant mutation types. *IKZF1* belongs to the IKAROS family of transcription factors, and its deletion/mutation frequently affects acute lymphoblastic leukemia. Clinically, two patients with mutated *IKZF1*, *PTPN11*, and *SF3B1* exhibited the same aggressive clinical course and showed primary resistance to chemotherapy [37]. *IKZF1* variants with whole-gene or exon deletions and novel fusions have been found, and patients with these variants developed lymphoid BC after 3 to 23 months of treatment with imatinib [10].

Wu et al. found that mutations in *KIT*, *CUX1*, and *GATA2* may play a role in TKI intolerance. These genes are related to myelodysplastic syndrome, and mutations in these genes affect hematopoiesis, especially in relation to cytopenia, which is associated with intolerance [38].

Using the genomic DNA of 91 patients with CML collected at the time of diagnosis and at other time points, Roche-Lestienne et al. examined four cancer genes, *TET2*, *IDH1*, *IDH2*, and *ASXL1*, which had been commonly reported in *BCR::ABL*1-negative hematological cancers. One of the 91 patients had *TET2* mutations at the time of diagnosis, and this patient transformed to a myeloblastic crisis 4 months later. No mutations were detected in *IDH1* or *IDH2* at the time of diagnosis, indicating that *IDH1/2* variants were rare at that time. In contrast, *ASXL1* frameshift and nonsense variants were found in 8 (8.8%) of the 91 patients at the time of diagnosis. However, the prognostic significance of mutated *ASXL1* remained unclear in this study. Of the 8 patients with mutated *ASXL1*, 3 patients achieved an MMR, and the other 5 patients had primary or acquired imatinib resistance. Three of the five patients with resistance had both *ASXL1* and *BCR::ABL1* kinase domain mutations. Overall, this initial study of only four genes revealed that variants were present in 9.9% of the patients with chronic-phase CML at the time of diagnosis [39].

The following Table 1 summarizes the characteristics and clinical relevance of each type of genomic abnormality in CML.

## 4. CML Drugs Expected to Be Used in the Future, Asciminib

Both first- and second-generation TKIs have improved the prognosis of patients with CML. However, CML cells can acquire resistance over the course of TKI treatments due to *BCR::ABL* mutations that lead to the replacement of amino acids in the *ABL* tyrosine kinase domain of *BCR::ABL*; these replacements change the structure of BCR::ABL, preventing TKIs from binding to ABL. Classical TKIs target the BCR::ABL catalytic ATP-binding site and reduce tyrosine kinase activity. Several *BCR::ABL* mutations have been reported, and second-generation TKIs are effective against most of them, except for the T315I mutation located in the middle of the ATP-binding site. Only ponatinib is effective against the T315I mutation. New alternatives are needed for patients with resistance or intolerance to ATP-binding TKIs, and for patients in whom the treatment goals are not achieved with ATP-binding TKIs. As mentioned above, there are a variety of resistance mechanisms in CML, and treatments are being developed to overcome each of them.

Unlike existing TKIs that bind to the ATP site of BCR::ABL1, asciminib (ABL001), an inhibitor specifically targeting the ABL myristoyl pocket (STAMP), has a novel mechanism of action that renders BCR::ABL1 inactive by allosterically binding to the myristoyl site of BCR::ABL1, altering the conformation of the ATP-binding domain [40] (Figure 3). Under normal conditions, ABL1 activity is autoregulated by the binding of the myristoylated N-terminus to the myristoyl pocket of the kinase domain. In CML, *ABL1* kinase is constitutively activated as a result of the loss of regulatory function due to the formation of the *BCR::ABL1* fusion oncoprotein. Asciminib binds to the myristoyl pocket of the ABL1 kinase domain, inducing an inactive conformation, and thereby inhibiting kinase activity [41]. Asciminib has had success in overcoming the resistance conferred by kinase domain mutations, even in patients with T315I treated by combination therapy. Of note, asciminib has less toxicity than other TKIs because of its high target specificity [42]. Although asciminib-resistant mutations have been predicted, they have not been identified to confer clinical resistance to ATP-competitive TKIs. Current clinical guidelines do not specify any mutations for which asciminib would be contraindicated, but such mutations will likely be added in the future with the availability of further clinical data. Due to different mechanisms of action on the same kinase, the combination of the other approved TKI with asciminib has been explored with the aim of reducing the possible appearance of BCR::ABL1 mutant clones. In vitro studies have shown that primary CML patient cells treated with asciminib and ponatinib had decreased BCR::ABL1 activity and colony formation when compared to those treated with asciminib monotherapy [43]. Asciminib and second-generation catalytic inhibitors have similar cellular potencies but distinct patterns of resistance mutations, e.g., there were no mutations in common for resistance to asciminib and nilotinib, which is a second-generation ATP-binding TKI [40]. In that study, rapid tumor regression was seen with single-agent asciminib or nilotinib treatment in a mouse xenograft model. However, resistance emerged with asciminib due to P223S and A337V mutations, and with nilotinib due to T315I mutations. In contrast, sustained tumor regression was observed with a combination treatment of asciminib and nilotinib [40]. Based on these results, a clinical trial was conducted to investigate the use of asciminib in patients with chronic-phase CML. Because asciminib is a relatively new TKI, further research and clinical trials are required to provide long-term therapeutic data and further insights into the role of *BCR::ABL1* kinase domain mutations in asciminib resistance.

A phase 1 open-label study (NCT02081378) was performed to determine the maximum tolerated dose and the recommended dose of asciminib. In total, 141 patients with chronic-phase CML and 9 patients with accelerated-phase CML who showed resistance to or unacceptable side effects from the use of at least two previous ATP-competitive TKIs were enrolled. The median follow-up was 14 months. Although 105 (70%) of the 150 patients had received at least three TKIs, among the patients with chronic-phase CML, 34 (92%) patients with a hematologic relapse had a CHR, and 31 (54%) without a CCyR at baseline had a CCyR. An MMR was achieved or maintained by 12 months in 48% of the patients who could be evaluated, including 8 (57%) of the 14 patients who were deemed to have resistance to or unacceptable side effects from ponatinib. An MMR was achieved or maintained by 12 months in 5 (28%) patients with a T315I mutation at baseline. The maximum tolerated dose of asciminib was not reached. Dose-limiting toxic effects included asymptomatic elevations in the lipase level and clinical pancreatitis. Common adverse events included fatigue, headache, arthralgia, hypertension, and thrombocytopenia, of which 92% were of grade 1 or 2. This study demonstrated that asciminib was effective in heavily pretreated CML patients who showed resistance to or unacceptable side effects from TKIs, including patients in whom ponatinib had failed, and those with a T315I mutation [44].

As a result of these findings, a phase 3 trial (NCT03106779) was conducted to verify whether asciminib may have better efficacy than bosutinib after second-line treatment. In this phase 3 open-label study, patients with chronic-phase CML who were previously treated with two or more TKIs were randomized (2:1) to receive either asciminib or bosutinib. The randomization was stratified by the major cytogenetic response status at baseline. The primary endpoint was the MMR rate at 24 weeks with asciminib vs. bosutinib. In total, 233 patients were randomized to receive asciminib (40 mg) twice daily (*N* = 157) or bosutinib (500 mg) once daily (*N* = 76). The median follow-up period was 14.9 months. The MMR rate at 24 weeks was 25.5% with asciminib and 13.2% with bosutinib. The difference in the MMR rate between treatment arms, after adjusting for the major cytogenetic response at baseline, was 12.2% (95% confidence interval (CI): 2.19 to 22.3), which was statistically significant (two-sided *p* = 0.029). Grade 3 or higher adverse events occurred in 50.6% and 60.5% of the patients receiving asciminib and bosutinib, respectively [45]. In addition, in a subsequent analysis after a median follow-up of 2.3 years, asciminib continued to demonstrate superior efficacy and better safety and tolerability than bosutinib. The MMR rate at week 96 (key secondary endpoint) was 37.6% with asciminib and 15.8% with bosutinib; the MMR rate difference between the arms, after adjusting for the major cytogenetic response at baseline, was 21.7% (95% CI: 10.53 to 32.95; two-sided *p* = 0.001). Fewer adverse events of grade 3 and higher (56.4% vs. 68.4%) and adverse events leading to treatment discontinuation (7.7% vs. 26.3%) occurred with asciminib than with bosutinib [46].

In the NCT03106779 study, the patients completed questionnaires to assess their CML symptoms and their interference with daily life (M.D. Anderson Symptom Inventory-CML (MDASI-CML), general health-related quality of life (HRQOL), five-level EQ-5D (EQ-5D-5L), Patient Global Impression of Change-CML (PGIC-CML), and Work Productivity and Activity Impairment Questionnaire-CML (WPAI-CML)). The HRQOL reported by patients is key to understanding the benefits and impacts of treatment on their lives and is becoming increasingly important as the life expectancy of patients with chronic-phase CML becomes longer and they require long-term treatment. The CML symptoms and HRQOL remained stable over 48 weeks of treatment with asciminib, with a general trend for a decrease in CML symptom severity, particularly for fatigue, and an improvement in the HRQOL [47].

Although asciminib was compared to bosutinib in the NCT03106779 study, the treatment recommendations for chronic-phase CML patients in whom two or more lines of treatment had failed were still unclear, partly due to the paucity of head-to-head trials evaluating TKIs (ponatinib, nilotinib, and dasatinib). Thus, matching-adjusted indirect comparisons (MAICs) were conducted to compare asciminib with competing TKIs as the third or later line of treatment in chronic-phase CML. Individual patient-level data for asciminib (ASCEMBL; follow-up: ≥48 weeks) and published aggregate data for comparator TKIs (ponatinib, nilotinib, and dasatinib) informed the analyses. The MMR, CCyR, and time to treatment discontinuation were assessed, where feasible. In this study, asciminib was associated with statistically significant improvements in the MMR by 6 months (relative risk [RR]: 1.55; 95% CI: 1.02–2.36) and 12 months (RR: 1.48; 95% CI: 1.03–2.14) when compared to ponatinib. Similar results were seen for the CCyR by 6 months (RR: 1.11; 95% CI: 0.81–1.52) and 12 months (RR: 0.97; 95% CI: 0.73–1.28) when compared to ponatinib. Asciminib was associated with improvements in the MMR by 6 months (RR 1.52; 95% CI: 0.66–3.53) when compared to dasatinib, but the CIs overlapped. Asciminib was associated with statistically significant improvements in the CCyR by 6 months (RR: 3.57; 95% CI: 1.42–8.98) and 12 months (RR: 2.03; 95% CI: 1.12–3.67) when compared to nilotinib/dasatinib. The median time to treatment discontinuation was not reached for asciminib in ASCEMBL. However, post-adjustment asciminib implied a prolonged time to treatment discontinuation when compared to nilotinib and dasatinib, but not when compared to ponatinib. These analyses demonstrated that asciminib leads to favorable outcomes when compared to competing TKIs, highlighting its therapeutic potential as a third- or later-line TKI treatment in chronic-phase CML [48].

Preclinical evidence indicates that the bone marrow microenvironment provides a protective niche for leukemic stem cells, allowing them to evade the effects of BCR::ABL1 tyrosine kinase inhibitors (TKIs). Ruxolitinib, a JAK2 inhibitor currently approved for myelofibrosis and polycythemia vera, demonstrated promising results in combination with TKI in increasing TKI-induced apoptosis [49]. Its mechanism of action is associated with the direct inhibition of JAK signaling but is also linked with enhancing MHC molecule expression, making CML cells more visible to the immune system [50]. The combination approach of ruxolitinib with TKIs is currently under investigation in different clinical trials.

Decitabine (DNA hypomethylating agents) has been tested as both first- and second-line treatments in CML patients. One hundred and thirty naïve CML patients were treated with escalating doses of decitabine, having achieved hematological responses in a significant number, but at the cost of prolonged myelosuppression [51].

In CML patients, particularly in those classified as high risk by the Sokal score, the expression of the immune checkpoint proteins PD-L1 and PD-1 has been observed. Thus, the upregulation of PD-L1 is considered an immunological escape mechanism for CML cells [52]. These data suggest that targeting the PD-1/PD-L1 pathway may be an effective strategy for eliminating CML cells.

Recent in vitro studies have shown increased phosphorylation of HSP90β serine 226 in patients who do not respond to TKIs. This site is known to be phosphorylated by Casein Kinase 2 (CK2), and CK2 has also been associated with CML resistance to imatinib. A CK2 inhibitor, CX-4945, induced the cell death of CML cells in both parental and resistant cell lines [53].

## 5. Conclusions

NGS studies have confirmed the role of mutations in cancer-related genes in CML. Since the co-occurrence of *BCR::ABL1* kinase domain mutations and cancer gene mutations is found in a high proportion of patients with TKI resistance, we should not consider resistance to be either *BCR::ABL1*-dependent or *BCR::ABL1*-independent; rather, resistance mechanisms may act interdependently to drive disease progression. As genetic analyses become increasingly affordable and available, they will be important tools for gaining a deeper understanding of the pathophysiology of CML. In future studies, larger patient cohorts are needed. In addition, pooled analyses of various studies should address whether additional genetic analyses could guide risk-adapted therapy and lead to a final cure for CML. Further clarification of the *BCR::ABL1*-independent resistance mechanism and the development of new drugs according to the resistance mechanism is also expected.

Asciminib showed significant efficacy with an acceptable safety profile as a later-line treatment for chronic-phase CML, with or without the T315I mutation. In addition, preclinical data indicated synergistic effects between asciminib and conventional TKIs in resistant CML. As such, a combination therapy of asciminib with conventional TKIs appears to have potential for clinical application in the future. In addition, asciminib monotherapy as a first-line treatment for newly diagnosed chronic-phase CML should be studied in the future.

## Figures and Tables

**Figure 1 ijms-24-13806-f001:**
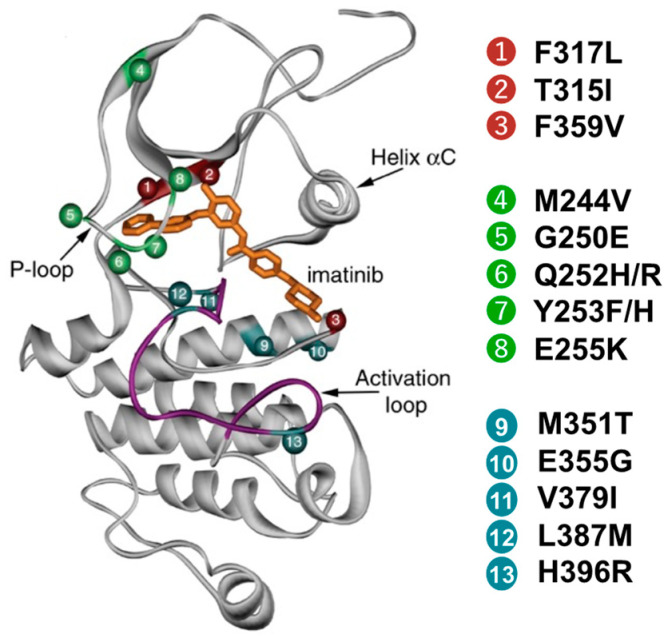
Mutations conferring TKI resistance in the ABL kinase domain.

**Figure 2 ijms-24-13806-f002:**
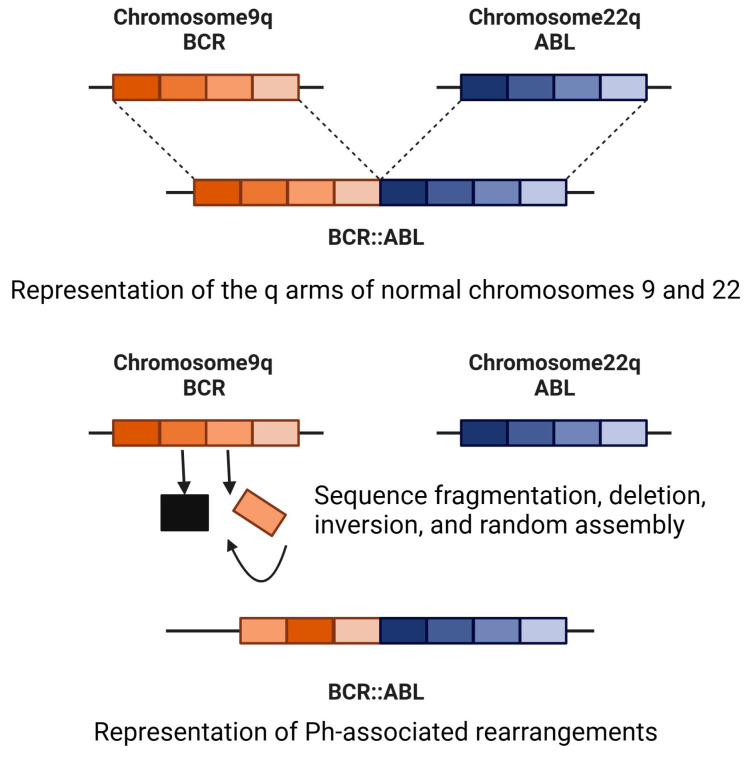
Rearrangements associated with the formation of the Ph chromosome.

**Figure 3 ijms-24-13806-f003:**
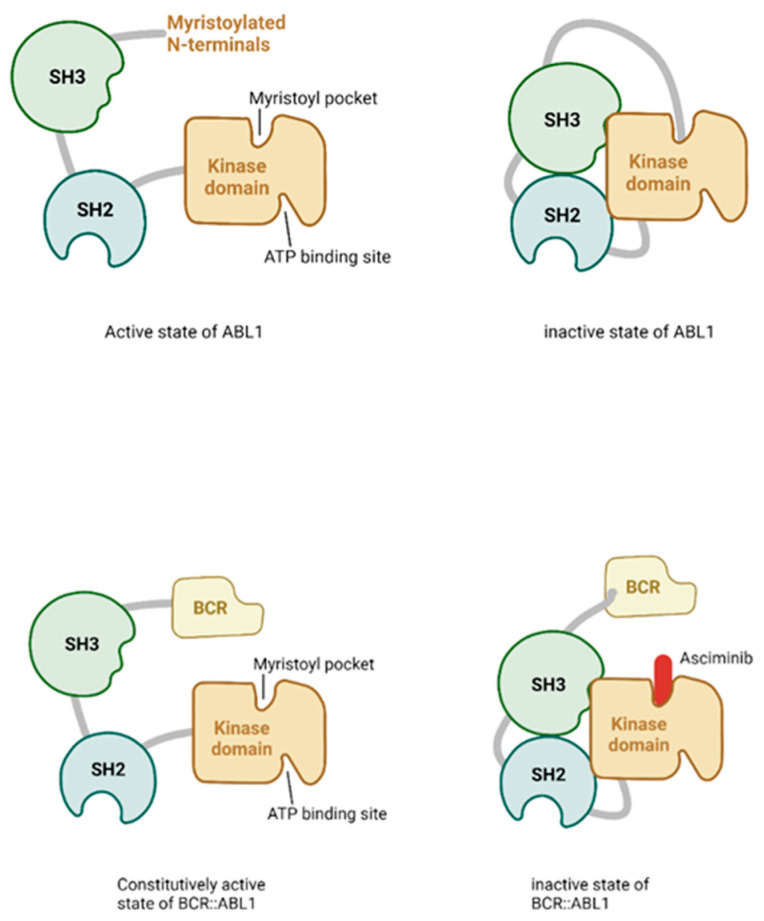
The BCR::ABL1-kinase domain and mode of action of asciminib.

**Table 1 ijms-24-13806-t001:** Summary of the genetic landscape of CML.

Genomic Abnormality	Mutation Characteristics/Clinical Relevance
Ph-associated rearrangements	Segments of chromosomes 9 and 22 become fragmented during the formation of the Ph chromosome, which results in the loss, inversion, and random reassembly of sequences that generate novel fusions. Associated with a poor outcome.
*ASXL1*	Leads to transcriptional abnormalities due to abnormal chromatin modifications. Most frequently mutated gene at the time of diagnosis in patients with a poor outcome. Longer time to blast crisis (21 months).
*RUNX1*	Encodes a regulator of hematopoietic cell differentiation. Contributes to the downregulation of DNA repair machinery and promotes genomic instability. Frequently mutated. Not always associated with a poor outcome.
*BCOR BCORL1*	In leukemia, BCOR and BCORL1 mutations disrupt the repressive function of PRC1.1 on target genes, resulting in epigenetic reprogramming and activation of aberrant cell signaling programs that mediate treatment resistance. Linked to advanced disease and clinical resistance to BCR::ABL-targeted inhibitors.
*EZH2*	Its role in CML remains unclear. Upregulation of EZH2 can result in a form of myeloproliferative disease. Rare.
*IKZF1*	Belongs to the IKAROS family of transcription factors. Frequently mutated. Most frequently detected in lymphoid blast crises.
*CUX1*, *KIT*, *GATA2*	Related to myelodysplastic syndrome. Affects hematopoiesis, especially in relation to cytopenia. Plays a role in TKI intolerance.
*IDH1* (*R132H*)	Leads to transcriptional abnormalities of related genes due to abnormal DNA methylation. Rare.

## Data Availability

Data sharing is not applicable.

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
