# Peer review of "Genetic Landscape of Chronic Myeloid Leukemia and a Novel Targeted Drug for Overcoming Resistance"

_ijms, 2023, doi:10.3390/ijms241813806_

Round 1

Reviewer 1 Report

The review is pretty brief and does not delve very deeply into the major questions regarding the current treatment of CML. It would benefit from a more in-depth look at the mechanisms of resistance.

Figures 1 and 2 are not really new and have been discussed extensively before.

The genetic landscape of CML in Table 1 needs to relate to the mechanisms of resistance and the newer therapies emerging in CML.

The authors would benefit from discussing how the newer agents prevent or or overcome drug resistance.

Author Response

Reply) 

Attached is the file for these precious comments.

Reviewer 2 Report

In the begining of page 2 description of disease stadia are not clear. On the same page in reporting of trials descriptions are not always clear: which medicine in comparison was better? It would be nice to cite the other attempts on overcoming the refractorines at least in vitro eg. work of Mirovsky O et al." Inhibition of casein kinase 2 induces cell death... Plos One May4, 2023.

Author Response

(The authors gave the same response as above.)
